# Nature-Based Therapy in Individuals with Mental Health Disorders, with a Focus on Mental Well-Being and Connectedness to Nature—A Pilot Study

**DOI:** 10.3390/ijerph20032167

**Published:** 2023-01-25

**Authors:** Lilly Joschko, Anna María Pálsdóttir, Patrik Grahn, Maximilian Hinse

**Affiliations:** 1Department of People and Society, The Swedish University of Agricultural Sciences, 234 22 Lomma, Sweden; 2Institute of Social Medicine, Epidemiology and Health Economics, Charité-Universitätsmedizin Berlin, Charitéplatz 1, 10117 Berlin, Germany

**Keywords:** garden therapy, psychosomatic patients, young adults, depression, stational care at hospital, effectiveness

## Abstract

In times of social and ecological crises, such as COVID-19 with lockdowns and implementing the impact of climate change, mental health degrades. Being outdoors in nature can be health-promoting, can decrease depression, and increase mental well-being. This pilot study investigated the relationships between nature-based therapy, mental health, and individuals’ connectedness to nature. We hypothesize that nature-based therapy has a positive impact on individual mental health and connectedness to nature. A mixed-method approach was used to evaluate the effectiveness of nature-based therapy for young psychosomatic patients. The results demonstrated improvements in mental well-being and connectedness to nature through therapy. Additionally, depression scores decreased. Patients reported the importance of the therapist setting the space, the supportive environment, the poems that fostered the nature connection, improvement at the soul level, and overall doing something meaningful. Every patient experienced nature-based therapy as effective. To conclude, the study gives a first insight into the processes of nature-based therapy in the German population at work and the effectiveness of nature-based therapy. Further questions, e.g., season effects, longitudinal effects, and whether patients with low connectedness to nature gain more out of the intervention remain unanswered.

## 1. Introduction

In times of neo-liberalized global capitalism and global warming [1,2], we might face sociological and ecological crises, such as COVID-19, more often. During the COVID-19 pandemic, green and blue outdoor environments (gardens, parks, and water areas) were demonstrated to have beneficial effects on mental health [3]. Having no access to green and blue spaces increases symptoms of mental health disorders (e.g., depression and anxiety) during lockdown [3]. In particular, a strict lockdown severity significantly affected mental health, while contact with nature helped people to cope with these impacts [3]. Moreover, individuals with higher connectedness to nature in times of a pandemic tend to adapt easier and faster to behavioral changes and therefore, respond to the crisis better [4]. Hence, nature-based therapy could help to cope with mental stressors in future pandemics.

The World Health Organization (WHO) reports that approximately 280 million people are yearly affected by depression [5]. Depression is characterized by a “[…] depressed mood (feeling sad, irritable, empty) or a loss of pleasure or interest in activities […] poor concentration, feelings of excessive guilt or low self-worth, hopelessness about the future, thoughts about death or suicide, disrupted sleep, changes in appetite or weight, and feeling especially tired or low in energy” [5]. Additional bodily symptoms (e.g., pain, fatigue, weakness) may appear, which are not due to another medical condition [5]. There is a gender difference in the diagnosis. Women are approximately twice as likely to experience depression as men [6,7]. Furthermore, the WHO states that depression and anxiety have a significant economic impact. The yearly estimated cost to the global economy is USD 1 trillion in lost productivity because of unemployment [8].

The prevalence of depression in Germany is approximately 15.7% (2017), with increasing tendencies over the past years [6]. Costs related to depression in Germany are calculated at approximately €3000–5000 per diagnosed patient for total healthcare costs [9]. Older individuals appear to cope better with the demands of depression and are less influenced by it [10,11].

Depression and anxiety also affect psychosomatic patients. Indeed, psychosomatic medicine is based on the biopsychosocial model and explores the connections between social and physical external contextual factors and how they affect patients behaviorally, mentally, and biologically [5,12,13]. Psychosomatic medicine focuses on diagnoses where mental, behavioral, and somatic processes together affect medical outcomes, and involve different specialties, such as psychiatry, psychology, sociology, occupational medicine, neurology, internal medicine, and psychoneuroimmunology [14]. Germany has long worked from a biopsychosocial perspective and with psychosomatic approaches [15,16]. A model for psychosomatic medicine was developed that has conceptualized and integrated psychotherapeutic methods in clinical practice in a way to explicate psychosomatic medicine in everyday practice in health and the healthcare system. Several diseases are believed to stem from stress and strain in everyday life, which can be treated with a psychosomatic approach, where stress relief is an important factor in treatments [15]. Due to their complex disorders, it is difficult to diagnose patients who often have a long period of suffering; moreover, it is expensive for the healthcare system because of the ineffective treatment before the patients receive the treatment they really need. Consequently, the clinical disorder in psychosomatic patients is complex, requiring a holistic approach as a treatment.

Therapy guidelines for depression in Germany include pharmacotherapy, psychotherapy, and non-pharmacological somatic therapeutic interventions, such as electroconvulsive therapy, awake therapy, light therapy, physical training, repetitive transcranial magnetic stimulation, and vagus nerve stimulation [17]. Additionally, there are complementary treatment methods that are used as therapeutic interventions. One of these complementary treatment methods is nature-based therapy [18,19].

Nature-based therapy, which is based on the supportive environment theory, has an interdisciplinary approach that is biopsychosocial [20,21,22]. It is therefore well suited to treat psychosomatic diseases. Nature-based therapy is located and conducted outdoors [23], where the outdoor setting is of importance. The setup of the place has a tremendous influence on the participants and the therapy outcome [24,25]. The accessibility of the natural environment was highly important to promote a feeling of freedom and means for change in clients in elderly care [24]. For clients with stress-related mental health disorders (such as depression and burnout), it was most important to have refuge (a secluded place) and serene (peaceful, silent, safe, and secure) spaces for the recovery process [26].

Overall, natural environments can have a positive impact on health and well-being [27,28,29]. Being outdoors in nature can be, in general, health-promoting [28]. Furthermore, being outdoors can stimulate all the senses [23,30,31], increase the feeling of freedom and have a positive effect on the immune system [31]. Moreover, light and air can increase reflection and self-regulation for patients with depression [31,32]. A close connection with the seasons and their change might serve as a mirror to oneself and it is suggested to promote self-acceptance and self-love [31,32]. More specifically, nature-based therapy (including nature-based rehabilitation and horticultural therapy) has been demonstrated to positively affect individuals with mental health disorders [26,33,34,35,36,37,38,39]. The research found that nature-based therapy improves patients’ self-esteem and decreases depression [40]; improves motivation and social interaction [41]; reduces healthcare consumption [39]; increases mental well-being, and engagement, and can give participants a sense of meaningfulness [21,35]. Overall, the potential of nature-based therapy is highly promising.

The overall recovery process in nature-based therapy might be influenced by the patients’ connectedness to nature. Connectedness to nature is defined as how strongly individuals feel they belong to nature and their individual emotional and cognitive beliefs about feeling related to nature [42,43,44]. Mayer and Frantz [42] suggest that connectedness to nature is an essential predictor of subjective mental well-being. Therefore, they invented the connectedness to nature scale (CNS) which is “[…] a measure designed to tap an individual’s affective, experiential connection to nature” [42] (p. 504). Choe, Jorgensen, and Sheffield [45] found that being in a natural environment, in particular, improves individuals’ connectedness to nature. Other studies investigated the correlation between connectedness to nature and mental well-being. Specifically, Cervinka, Röderer, and Hefler [45] found in their study ‘Are nature lovers happy?’ that psychological well-being is robustly correlated with connectedness to nature in healthy individuals. Indeed, further research found that connectedness to nature is positively associated with a subjective perception of happiness, well-being, physical and mental health, and life satisfaction [42,44,45].

The question arises: which underlying mechanisms drive the relationship between nature connectedness and mental well-being? Research in Japan demonstrated that place attachment had a positive and significant mediating effect on this association [46]. The relationship between nature connectedness and place attachment and between place attachment and individuals’ well-being was direct and significant. Consequently, their findings suggest that higher levels of well-being associated with nature connectedness are due to the sense of attachment to a place that nature provides [46]. Palsdottir et al. [21] found that the first part of nature-based therapeutic activities was about the patients finding a place in the garden (which they called their nature-place) where they could feel safe and de-stressed. For the therapy to work, this process was crucial. Grahn et al. suggest that this type of place attachment is similar to Bowlby and Ainsworth’s theory on human attachment [47,48,49,50]. The theory they put forward—the Calm and Connection Theory—is about archaic basic neurological systems being activated in natural areas that provide peace and security. Grahn et al. [50] propose that place attachment to natural environments (which can be described as nature connectedness) provides a possible role for the human oxytocinergic system to function as a physiological mediator for positive and health-promoting effects. Oxytocin promotes various types of social interaction and bonding and results in stress-reducing and healing effects (e.g., anti-inflammatory). Oxytocin is released, according to the theory, in connection with the attachment or bonding to certain natural places, developing in particularly safe and attractive places. Upon release, levels of fear and stress decrease, while levels of trust and well-being as well as health-promoting effects increase. Furthermore, the ability to develop coping skills or psychological development can also be promoted [50].

Our aim with this study was to investigate the relationships between nature-based therapy, mental health, and individuals’ connectedness to nature. Our hypotheses are as follows:Nature-based therapy has a positive impact on individuals’ mental health.Nature-based therapy fosters individuals’ connectedness to nature.There is a difference in the improvement of psychological well-being in patients with low vs. high connectedness to nature after nature-based therapy.

Does connectedness to nature improve mental well-being?

## 2. Materials and Methods

This study was conducted at the Gemeinschaftskrankenhaus Havelhöhe (GKH) hospital, Berlin, in the psychosomatic ward as an observational real-world data, prospective, cohort study without a control group. The study was part of a larger study evaluating the anthroposophical complex number in the DRG system (study “EVAL26”, registered under DRKS00020547) [51]. The study was reviewed and approved by the Ethics Committee of the Charité-Universitätsmedizin Berlin (EA2/089/19 dated 31 September 2019). All study participants gave their written informed consent to participate in the study. The study took place in compliance with professional regulations, the Declaration of Helsinki, and the recommendations of the ICH Guideline for Good Clinical Practice [52,53]. The authors confirm that all ongoing and related trials for this intervention are registered.

Patients of the ward, young adults (from 18 to 27 years), participated in nature-based therapy as a standard practice during their stay. Inclusion criteria were: minimum age of 18 years, ability and consent to participate in the study, and participation in nature-based therapy. Exclusion criteria were lack of understanding of the German language and severe acute or chronic illness, making participation or completion of the questionnaire impossible.

### 2.1. Design

During the period from May until July, the study took place at the GKH. The study was the subject of the Master’s thesis by the first author at the Swedish University of Agricultural Sciences (SLU). Furthermore, the study uses a mixed methodology approach [54].

The qualitative methods using a descriptive exploratory methodology were open questions in the questionnaire and participant observations [55,56,57]. In this case, the first author was present continuously and involved in the nature-based therapy in the garden at GKH and the transdisciplinary team meetings with the other therapists during the time of the study. This was accomplished to obtain an insider’s perspective by becoming part of the intervention and natural setting of the therapy. The role of the researcher was to explore and inspect to better understand how, and possibly, why, the connectedness to nature and mental well-being was influencing the mental health of the patients. The researcher also offered possibilities to engage in garden activities, shared garden knowledge, had individual talks with the patients, carried out observations, and took care of the data collection; consequently, the researcher had more insight into the different and complex processes at work during the intervention. The researcher is a trained and educated speech and language therapist and has previous experience in providing therapy. Thus, she knew how to approach patients and possessed background knowledge of therapeutic interventions and the patient-therapist relationship.

The quantitative part employs the positivist paradigm as its guiding methodological framework to evaluate and examine the outcome of the nature-based therapy program at the GKH hospital, Germany. Data were collected over a period of 3 months. Standardized questionnaires were answered before and after (as pre- and post-measurement) the participants’ 4-week stay at the hospital.

The quantitative evaluation of the treatment measured change of time at two measurement points: baseline (pre-) and post-testing design (see Figure 1). Since the stay at the hospital is not always six weeks, the decision was made to collect data only after the fourth week for each participant. This was to ensure comparability of the data among the participants. Thus, each person who received nature-based therapy filled in the questionnaire on two occasions: the first day of their nature-based therapy as pre-testing (before the intervention) and the second time, after the 4th week of the nature-based therapy as post-testing (after the intervention).

The integrative therapy method at the GKH hospital, in which all patients participate, includes nature-based therapy (3 sessions, each comprising 60 min per week); psychotherapy with the integrative schema therapy approach [58] (1 session comprising 60 min per week), psychoeducation [59] (3 sessions, each comprising 50 min per week), music therapy (2 sessions, each comprising 50 min per week), painting/sculpting therapy (3 sessions, each comprising 60 min per week), animal-assisted intervention (4 sessions, each comprising 75 min per week). See Figure 1. It is recognized that the integrative therapy method, including nature-based therapy, has a good impact on the health of patients. Nevertheless, this study focuses on nature-based therapy; therefore, the measurements chosen for this study focus on nature-based therapy.

### 2.2. Procedure

#### Nature-Based Therapy

The nature-based therapy program for the psychosomatic patients at the GKH takes place three times a week, with one hour allocated for each session. All sessions take place outside, either in the garden (twice a week) or in the forest (once a week). The therapy garden of the GKH is located on the hospital grounds at a distance of about 150 m from the patients’ house. The hospital is partly surrounded by forest in which the forest therapy units take place. The nature-based therapy takes place in a group setting and comprises a maximum of 12 psychosomatic patients. The group is constantly changing: each week, approximately two participants start their stay at the hospital and two leave the group by ending their stay.

Each nature-based therapy session in the garden starts in the back corner of the garden, hidden underneath huge chestnut trees with everyone sitting on logs in a circle (see Figure 2). The trained nature-based therapist quotes a self-written poem matching the topics, which are specific to the patients in the group. The specific issues were discussed in the therapist meetings held by the transdisciplinary team of caregivers. Following this, the nature-based therapist connects the poem’s content with the topics, including, why the patients are here and their struggles. The therapist also added his own wisdom, referring to his coping strategies and everyday uplifting positive thoughts. This is a way to address the patients’ struggles and foster an acceptance of destigmatizing the patients’ diagnoses and disorders. It is also a way to ensure that everyone feels seen, heard, and accepted as they are. Moreover, this approach supports and enhances practices of self-love.

Each nature-based therapy session in the forest starts with a walk all together from the patients’ house into the forest at the clinic area (approximately a 1.5 km walk). The session starts with an opening circle and poems—the same as in the garden. After, there can be group walks among nature with further poems and sharing rounds, land art tasks (e.g., finding and creating a home in nature where every individual feels safe, inviting the therapist or the other clients), etc. At the end is a gathering circle and walk back to the clinic house together.

Nature-based therapy often uses a tool called nature metaphors. They provide analogies and are a simple way to understand complex reasoning, enhance information processing, cognitive flexibility, and the ability to remember and recall information [60]. Metaphors teach specific skills, such as cognitive restructuring, cognitive rehearsal, and exposure in an engaging way [61]. Furthermore, the destigmatization of mental health disorders is necessary to seek and participate in mental health care [62]. In particular, combining metaphors, cognitive flexibility, and reflection on life issues through nature poems might reduce mental health stigma [63].

After the opening circle of the therapy session with a poem and wisdom from the therapist, possibilities to engage in gardening work are named by the therapist. The activities are used metaphorically as well. For example, grass that grows along the edges of the flower beds could be pulled out by hand. If someone’s topic were to set clear boundaries, this task would suit them. The participants are always informed that if they see a task that could be performed, and they feel like doing it, they are welcome to perform that task directly. The therapist closes the session under the trees, and everyone chooses a task. Nature-based therapy using the NMBC method (Nature, Mind, Body, Community spirit) [64,65] has the same approach, to some extent. Participants gather at the beginning and end of each session around a fire to share stories and experiences. According to this method, the participants are introduced to various nature experiences and stories about plants and animals, with the intention of opening up the participants’ awareness of nature. By gaining more excellent knowledge of and experiences in nature, place-attachment in a natural environment could be more easily built [46]. Through storytelling, the participants can also build social cohesion in the group [66], and it provides new concepts and paradigms for healthy behavior [67].

During the nature-based therapy session, the therapist does some gardening work as well. Additionally, he keeps an eye on the patients to see if everything is going well. The therapist is always there to answer questions and give advice regarding the tasks. Moreover, he asks the patients if everything is going well and offers the possibility to engage in talks and other tasks, too. At the end of the session, he asks the patients to clean up their space and return the tools they used. Thus, after the one-hour therapy session, everyone goes back together to the psychosomatic house in the clinic where they are staying. In general, the patients are welcome to spend their free time in the garden as well and carry out watering over the weekend when it is a hot day. However, good care is taken to ensure that the tasks are always freely chosen and are performed on a voluntary basis. After the stay of 4–6 weeks, each patient receives a letter from the therapist to take home. There, he or she quotes a poem that suits the feeling and the personal situation of the specific patient. Additionally, the therapist writes some personal words of support and well wishes for them to take home.

Overall, the intervention is an invitation that helped the patients to feel a connection with nature, to use nature as an illustration of transformation, and struggles, as well as a positive example. The analogy is that plants are seeded, grow, die, rest in the winter, and then grow again in the summer.

### 2.3. Outcomes

Qualitative data were collected via open questions in the questionnaire and participant observations, such as talks during the nature-based therapy with the researcher and the therapist. After each session, the researchers wrote down what they saw, how participants behaved, talks with the clients, little quotes of what they said and which activities they joined, and which poems were cited at the beginning of the session.

All quantitative outcomes were collected with standardized and validated questionnaires. The primary outcome is the subscale ‘mental well-being’ (German “psychologisches Wohlbefinden” (WOHL)), taken from the HEALTH-49 questionnaire [68]. Secondary outcomes are the connectedness to the nature scale (CNS) [42] and the Patients Health Questionnaire (PHQ-D) [69]. In the following sections, each questionnaire is described in detail.

#### Questionnaires

Demographic data, including age, gender, and diagnosis, were obtained during the baseline of the nature-based therapy. The self-rated questionnaires (baseline- and post-questionnaire) used for this research comprised four standardized, self-completed questionnaires and diagnostic instruments to measure the outcomes of nature-based therapy in terms of connectedness to nature, as well as some questions to evaluate the effectiveness of nature-based therapy as one part of the integrative approach.

WOHL. The subscale ‘mental well-being’ (WOHL) of the HEALTH-49 questionnaire, German version [68,70], was used. The test has been shown to meet the following Q criteria: practicability, dimensionality, reliability, validity, and sensitivity to change (ibid). It consists of five questions, using a 5-point Likert scale from 0 to 4. The higher the sum score, the better the mental health of the patients. The sum score range is from 0 to 20. Cut off is 1.821 and the critical difference is 0.628.

CNS. The Connectedness to nature scale (CNS) [42] comprises 13 questions, answered on a 7-point Likert scale. In this study, it is used to examine an individual’s connectedness to nature. The scale has been demonstrated to have good psychometric properties, such as validity and reliability. Higher scores on the scale indicated a higher connectedness to nature.

PHQ-D. The German version of the Patients Health Questionnaire (PHQ-D) [69] was used to classify the diagnoses of the psychosomatic patients. The PHQ-D is a valid, effective, and well-accepted scale. Specifically, it classifies various disorders. The subscales used in this study are for: depression (PHQ-9 [71]), somatoform disorders (PHQ-15 [72]), and stress. All three use a 4-point Likert scale. The depression subscale has 9 items and classifies depression with a total score from 0, indicating no depression, to 27, indicating severe depression. A score of five or more indicates mild depression, from 10 to 14 moderate, from 15 to 19 moderate-severe, and above 20 severe depression. The Somatoform disorder subscale has 13 items and uses a total score from 0 to 30. Scores of 5, 10, and 15 represent cutoff points for low, medium, and high somatic symptom severity in the PHQ-15. The stress subscale comprises ten items, with the scores ranging from 0 to 20. The higher the score, the more severe the impairment.

The participants’ self-assessment of the effectiveness of the nature-based therapy was evaluated with the post-questionnaire (for post-testing). It was created for this research to evaluate the effectiveness of nature-based therapy. For this study, eight questions were used to focus on the nature-based therapy that takes place at GKH. They are embedded in the overall “Questionnaire to evaluate the effectiveness of anthroposophical medical complex treatment from the patient’s perspective,” created by the ‘Working Group integrative and anthroposophical medicine,’ from the Institute of Social Medicine, Epidemiology, and Health Economics at the Charité-Universitätsmedizin, in Berlin. The two overall questions about the effectiveness of and satisfaction with nature-based therapy used a 5-pointLikert scale from very (5 points) to not effective (2 points), including not applicable (1 point). The other, more specific, six questions were answered by a 6-pointLikert scale from fully agree (6 points) to disagree (2 points) and not applicable (1 point). The questions focus on positive improvement on mood level, bodily level, soul level, on grievance, on contact with other people, and on coping with problems and illnesses. “Soul level” is a concept used in counseling [73], psychotherapy [74], and nursing [75], as a holistic concept of consciousness and being. Tucakovic [75] describes being as a function of the soul. A higher degree of soul level can be described as a higher function of a person’s being, presence, or consciousness. The questionnaire will also be used for further investigations for the research institute and the hospital, even after the data collection for this study is completed.

Attention was paid to using simple and clear language in the questionnaire following the guidelines of Statistics Sweden (2004) ‘Design your questions right’ [76]. Consideration was also taken to double the necessary pre- and post-questions and to not assess further unsuitable questions twice.

### 2.4. Statistical Analysis

Given that this was a pilot observational study, the sample size was not calculated. All results are considered exploratory. In addition to descriptive results, *t*-tests were performed for the before/after comparisons of the nature-based therapy outcomes. All outcomes were reported with pre- and post-nature-based therapy results with false discovery rate correction for multiple testing (q value).

Two linear mixed model analyses were performed for the main analysis. The first analysis included a base model with only the treatment effect as a predictor of mental well-being (WOHL subscale). The second model then included the secondary outcomes as predictors (CNS and PHQ subscales).

Both models included the participant’s ID as a random effect (39–41). Two models were used to better differentiate the effect of secondary outcomes (model 2) from the main effect (model 1). In order to investigate the influence of different covariates, such as depression or anxiety disorder, on the success of the AMT, these were included as covariates in the multivariate mixed effect analyses. Fit measures are reported as well (AIC, BIC, RMSE, Sigma, and ICC).

Analyses were conducted using the R Statistical language (version 4.1.0; R Core Team, Vienna, Austria, 2021) with RStudio Version 1.4.1717 on macOS 12.0.1, using the most recent versions of the R packages: tidyverse, lme4, ggstatsplot, ggeffects, sjPlot, gtsummary. A 5% significance level was set for the statistical analyses [77,78,79,80,81,82].

### 2.5. Qualitative Data Analysis

The explorative open questions answered by the patients in the questionnaire and researcher’s field notes (such as observations, quotes, talks with patients from the sessions, and talks and observations from the therapists) were collected and transcribed into one document. This document was treated as one type of data, although the open questions from the questionnaire were given more weight and used as direct citations in the discussion section. The first step was to collect all the material and sort it into different topics: open questions from the questionnaire, observations from the researcher, observations from the therapist, and notes from the notebook regarding poems and talks during the therapy. Thereafter, all the material was translated from German into English. Next was the stage of inductive coding which involved reading and re-reading the material, connecting the emerging topics and themes based on the thematic psychology approach [83]. The realistic psychology approach was the basis of making meaning out of the bits and pieces collected. A narrative analysis was then written from the researchers’ perspective to provide new meaning by synthesizing the experiences and sessions into a coherent whole [84]. The key meanings, themes, and ideas obtained from the qualitative data analysis were triangulated with the results of the quantitative analysis in the discussion. It is acknowledged that the researcher’s perspective is highly important and a lot of reflection is needed to observe as objectively as possible [55]. The narrative approach is also influenced by symbolic interactionism, which investigates how meanings are constructed by individuals (in this case, the patients) within their social and personal world [85]. An additional point of view is given to pluralistic knowledge, which means that we shift the paradigm toward multidimensional mechanisms that take place at the same time [86]. It is not about one factor that influenced the health of the patients; rather, it is about finding out the different mechanisms at work during nature-based therapy, as experienced by the patients.

## 3. Results

### 3.1. Sample Description

A total of 20 patients (18–27 years old) participated in the study. One person dropped out during the baseline collection, after giving their consent to participate in the study. Thereafter, 19 participants remained (see Figure 3).

Specifically, 16 female and three male patients took part in the baseline data collection (see Table 1). Their average age was 21 years. Not all patients concluded their four–six-week rehabilitation program because they broke the house and/or COVID-19 regulations. Consequently, three female participants did not take part in the post-testing, since they had to leave the hospital earlier.

All patients received a diagnosis of depression (mild–severe) with the German version of the patient’s health questionnaire (PHD-Q) [69]. Most patients (63%) were deemed to have a pronounced and severe depression (n = 12). Other additional diagnoses are somatoform disorders, insomnia, acrophobia, anxiety disorders, social phobia, post-traumatic stress disorder (PTSD), anorexia, obsessive-compulsive disorders, panic disorder, and bulimia. Patients in the psychosomatic clinic at the GKH can be admitted either by their general practitioner (resident physician) or by their own request.

### 3.2. Quantitative Results

To evaluate the effect of the nature-based therapy, the paired *t*-test was performed (results, see Table 2). The paired *t*-test tested the difference between the mean of the baseline and post-condition of mental well-being (WOHL) and for all the secondary outcomes. The paired *t*-test, testing the difference between the WOHL baseline (1.05) and WOHL post-testing (1.60; mean difference = 0.55), suggests that the effect is statistically significant and large (*p* < 0.001; Cohen’s d = −1.04, see Table 2). The difference of 0.55 measured here is below the critical difference of 0.68 that is given by the author for clinical relevance.

The difference between the CNS baseline (3.91) and mean CNS post-testing (4.37; mean difference = 0.45, *p* = 0.010) suggests that the effect is statistically significant and medium (Cohen’s d = −0.74, see Table 2).

The paired *t*-test, testing the difference in depression between the PHQ-9 baseline (16.94) and the PHQ-9 post-testing (12.94; mean difference = 3.31), suggests that the effect is statistically significant and medium (*p* = 0.044; Cohen’s d = 0.55, see Table 2). Even if the effect for depression is just statistically significant here and the mean score changes from moderate-severe at baseline to moderate in post-testing, the reduction of 3.31 points in the PHQ-9 is not a clinically relevant reduction in the depression scores.

PHQ scores for the subscales for stress and somatoform disorders are both non-significant differences with small and very small effects (see Table 2).

For our main analysis, the effect of nature-based therapy on mental well-being, we fitted two linear mixed models (estimated using REML and nloptwrap optimizer) to predict mental well-being (WOHL) with treatment in model 1. Both models included patients’ ID as a random effect. The total explanatory power of base model 1 predicting mental well-being only by treatment is substantial (conditional R^2^ = 0.61), and the part related to the fixed effects alone (marginal R^2^) is 0.21. The model’s intercept, corresponding to treatment = baseline, is at 1.05 (95% CI [0.80, 1.31], t(31) = 8.41, *p* < 0.001). The effect of treatment [post] within this base model 1 is statistically significant and positive (beta = 0.55, 95% CI [0.28, 0.82], t(31) = 4.12, *p* < 0.001; Std. beta = 0.92, 95% CI [0.46, 1.37]). As could also be seen from the pre/post comparisons of the measured outcomes, base model 1 of the liner mixed effect analysis reflects the result. The inpatient stay has a positive effect on the mental well-being of the patients, measured with the subscale WOHL of the HEALTH-49.

To distinguish the effect of nature-based therapy more specifically, we included the effect of connectedness to nature with the other secondary outcomes in our second linear mixed model predicting mental well-being (see Table 3 and Figure 4). While having patients’ ID as a random effect included (estimated using REML and nloptwrap optimizer), all others are fixed effects predicting mental well-being (WOHL) with treatment, CNS, PHQ-9, PHQ stress scale, and somatoform disorders (PHQ-15). The second model’s total explanatory power is substantial (conditional R^2^ = 0.74, Table 3), and the part related to the fixed effects alone (marginal R^2^) is 0.35. The model’s intercept, corresponding to treatment = baseline, CNS = 0, PHQD = 0, PHQS = 0, and PHQSO = 0, is at −0.21 (see Table 3). Within this second model, the effect of Treatment [post] is statistically significant and positive (beta = 0.46, *p* = 0.003; see Table 3). This effect suggests that treatment was effective in helping patients gain a higher mental well-being score in the WOHL subscale of HEALTH-49 during their inpatient stay in the GKH hospital (see Table 3). Patients’ connectedness to nature (beta = −0.13, *p* = 0.191), the depression score measured with the PHQ-9 (beta = −0.03, *p* = 0.265), and the stress scale from PHQ-D (beta = −0.04, *p* = 0.194) are statistically non-significant and negative. The effect of somatoform disorder (PHQ-15) is statistically non-significant and positive (beta = 0.00, *p* = 0.895). All secondary outcomes have no statistically significant effect on predicting mental well-being in this group of patients. Patients’ connectedness to nature increased during their stay in the hospital (see Table 2), but it has a non-significant and very small negative effect in predicting mental well-being. All secondary outcomes have no noteworthy correlations with all other outcomes, thus having no impact on the linear mixed effect analysis and additionally also no noticeable interactions. Despite all this, the secondary outcomes contribute to an overall improved second model (Chisq = 11.432, df = 4, *p* = 0.022, AIC = 54.691, BIC = 67.134, LogLik = −19.346).

The ’Evaluation of the nature-based therapy from the patient’s perspective’ was filled in by 15 participants. All participants experienced the nature-based therapy as effective (see Figure 5). Specifically, 86.66% experienced the nature-based therapy as satisfactory, while 53.32% witnessed the nature-based therapy as supportive, and self-reported an improvement in their condition. Furthermore, 40% experienced an improvement on a bodily level, 93.33% encountered enhancement on a mood level, and everyone experienced improvement on a soul level. Moreover, 46.66% felt an enrichment in their well-being, while 53.33% experienced that their contact with other people had improved. In addition, 86.66% encountered development in coping with problems and illnesses. Interestingly, in this part of the questionnaire, most individuals (87%) felt they had improved in coping with problems and illness through nature-based intervention (see Figure 5).

### 3.3. Qualitative Results

Patients described that they felt safe in the garden because of the environment and the therapist and researcher who was setting up the place. The therapist was described as being empathetic, accepting the patients as they were in the moment when they sought help, meeting them with unconditional valuation, and fostering their connection to themselves while fostering their nature connection. The therapist created an environment where everyone felt safe, seen, heard, inspired, encouraged to explore, and truly accepted as they were. It was not about performance; rather, it was about being. It was about re-connecting with oneself, i.e., connecting with one’s own feelings, learning self-acceptance, and self-love out of the numbness they experienced before. One client said, “Normally, I withdraw in nature. I felt connected, but now, it makes me realize that this blunt feeling is there.” There were no expectations about how they should be, what had to be included in the session, or what had to be accomplished.

Nature-based therapy was about the process of doing, exploring the slowness, re-connection to oneself, and feelings in general, but also joy. The patients could choose what tasks they wanted to do themselves, individually and at their own pace. This atmosphere of acceptance and openness was facilitated by the staff and the environment. The therapist said, “Through nature-based therapy, the patients learn acceptance with impermanence. Change in the garden and in the self.” Additionally, the therapist encouraged the patients at the beginning of each session to reflect on the topic of the poem. This was achieved indirectly—without a task or any demands. It was achieved with a high degree of freedom. It was achieved through inspiration from the poem and the experiences the therapist shared from his own life. The therapist’s calm attitude and the garden revealed fascination and recreation. The therapist acted as the door opener for the clients to look at nature and themselves in a more reflective, metaphorical, and accepting way, being in the here and now, mindful through linking nature with human behavior. For example, one poem was about walking through the thicket and bushwhack and mosquito infestation. Along the way, losing courage, getting lost, not knowing the way, back or forth, becoming afraid. However, in the end, arriving in passing. After quoting the poem, the therapist said, “It is a challenge to be here, but it is also an invitation to change; it can always be the next step to leave the thicket, to arrive. It is worth going further”. With these words, he heartened the patients; he was full of acceptance, courage, and love. He destigmatized their choice to go to the psychosomatic clinic because they felt lost.

The activity for the day was to sow seeds, which can be interpreted as inspiration for change—the next step to leave the thicket. The process of recovery and starting something new. The seeds of inspiration. This special interplay of so many layers of reinforced recovery was described by one patient as: “Especially important and formative for me were and are the talks at the beginning where many topics are conveyed, particularly empathetically and skillfully, which are sometimes general, but formulated so that everyone can take something for themselves. I was able to rethink many things and took away a lot of courage and confirmation. The honesty, attentiveness, empathy, courage that I always encountered have enriched me very much.”

The poems and the gardening work have been seen as repetitive to reflect on one’s own topics from psychotherapy, possibly directly making changes to one’s behavior. One patient wrote: “I particularly remember the silence after a poem, where everyone was completely with themselves for a moment.”

Through this holistic approach in the garden, the therapist fostered the participants’ connectedness to nature. Some had none; they never felt a connection to nature; and they never stopped to just look at nature, to be there, to use the time to reflect. Some were also afraid of being in nature because they did not feel safe outside at all. However, through the therapy, one patient told me that from now on she would take time for herself in nature. To sit and relax. Just watching the green space, doing nothing, and potentially thinking about the therapy and the garden at the hospital.

Many patients asked for a plant they could take home with them—to have something life-like from the clinic. A visible change, a trophy, a memory of lessons learned. To remember to integrate them into their everyday life. Patients often said, “The plant will remind me to take care of not only the plant but also myself.” It is a way to remember self-care, self-love, and self-acceptance. Furthermore, patients developed a curiosity and a new interest in plants. They approached the researcher and the therapist to ask about the names of the plants and used their limited online time to research the plants and their needs.

Overall, nature-based therapy helped patients to trust in change and the small changes that are within us—not directly visible. Taking plants or a tree pit from therapy, (which was connected to a poem and wisdom that everything we learn is stored within us) with them fostered their transfer effect. It allowed them to transfer lessons learned from the therapy into their everyday life at home. One patient wrote: “When I go home and I have the feeling that I didn’t learn anything, or I fall back into my old habits—the words of the therapist and caregivers come to use and I carry them with me—consciously or unconsciously. I created new rings as well, and they won’t disappear.”

## 4. Discussion

Our aim was to investigate the relationships between nature-based therapy, mental health, and individuals’ connectedness to nature. Specifically, the first hypothesis was to investigate if nature-based therapy has a positive impact on individuals’ mental health. The results from the WOHL support our hypothesis that there is a change over time in mental well-being in psychosomatic patients through nature-based therapy (see Table 2). Receiving nature-based therapy increased mental well-being significantly. Although not clinically relevant, the results also showed that self-reported symptoms of depression decreased significantly, as measured by the PHQ-9. The severity of the diagnoses of depression decreased: through their therapy program (including nature-based therapy), the depression scores decreased. During the baseline, 12 individuals had pronounced and severe depression. At the post-testing, seven individuals continued to have pronounced and heavy depression scores. These results indicate that nature-based therapy might decrease depression severity. On the whole, the following three scores: depressiveness, somatoform disorders, and stress decreased through the treatment. From baseline to post-testing, the mean for all three scores decreased. Thus, it is assumed that the therapy all patients received, including nature-based therapy, might have a positive effect on these three disorders. These changes might be due to the treatment the patients received in nature, but might also be a side effect from the other therapy sessions and activities that clients received. There was no evaluation of the overall complex treatment the clients received; therefore, there is no conclusion about which treatment helped whom the most. This could be further investigated. However, results from the participants’ self-assessment questionnaire on the effectiveness of nature-based therapy showed that they considered the therapy to be effective, not least regarding mood. This effectiveness of nature-based therapy goes in line with the findings from other nature-based intervention studies [21,36,38,39,87,88,89,90,91,92].

Our second hypothesis was to investigate if nature-based therapy has a positive impact on individuals’ connectedness to nature. The results from the connectedness to nature scale (CNS) support our hypothesis that there is a change over time regarding connectedness to nature in psychosomatic patients through nature-based therapy (see Table 2). Receiving nature-based therapy increased connectedness to nature significantly. The natural environment itself and the therapy together could have fostered this connectedness. The Calm and Connection Theory posits that environments and situations that trigger a release of oxytocin can lead to promoting connectedness to a natural environment. Studies have found that for nature-based therapy to work, participants first need to find a place they like, where they can find security and want to return to. They also need to find opportunities for meaningful activities where they feel they can accomplish something, feel satisfaction, and joy and that they can grow [20,21]. According to the Calm and Connection Theory, these conditions provide an opportunity for a release of oxytocin, which reduces stress levels, while levels of trust and well-being increase, which, in turn, promotes attachment, or connectedness, to the place [50]. According to the Supportive Environment Theory [21], nature-based therapy must include three supportive environments: a physical environment, a social environment, and a cultural environment. The cultural environment is conveyed through language and activities, as well as through symbols in the environment, for example, in the expression of the surrounding nature and how the garden is designed. The supportive environments should also include a gradient so that they support people with very poor coping resources to those with growing resources. Moreover, the therapy must be led by a person with great empathy and sensitivity to the participants’ needs [21]. If nature-based therapy is carried out with such conditions, health-promoting effects, social ability, and the ability to develop coping skills will be promoted [50]. Studies also show that in such nature-based therapy settings, patients experienced nature-based rehabilitation as a meaningful occupation [67] and the place as a restorative environment [26]. The restorative environment was experienced as very important for their recovery process. This was also true for the offered activities, which were adaptable to the individual needs of the patients. They could be passively engaged (inner involvement) or actively outgoing [26]. A similar structure was offered in the nature-based therapy program that the current study investigated.

Themes from the qualitative interviews showed that the participants considered that the nature-based activities and the setting itself caused health-promoting effects. Different gardening activities were offered, and everyone was allowed to be themselves, to show their inner self—be authentic and engage in the activity they liked. The therapy program might have fostered the individuals’ meaningfulness and connection. One patient wrote what she liked in the nature-based therapy was, “The feeling of doing something and above all doing something meaningful and good, which helps me and nature”.

“Meaningfulness, in contrast to depression, is understood as a developmental motive, referring to a human’s need of being in the world and experiencing a sense of purpose in life” [45] (p. 385).

Finding meaning in life is important for one’s well-being and might be one of the most important aspects of living [93], and it is the core of the concept of salutogenesis [94,95]. Garden activities can give meaning [21,35,88,94,95,96]. This might have an evolutionary explanation [97], since back in the day, gardening was necessary for survival, as it provided food for one’s social group or family. The results from the current study on improvements in well-being (see Table 2) and the effectiveness of the nature-based therapy (see Figure 5) support the findings from the literature. The majority of patients indicated that their contact with other people has improved during the nature-based therapy (see Figure 5). The explanation for an increased connectedness might be the therapy group setting, where people feel connected as a group. They do activities together, knowing they are not alone with their disorder. Feeling accepted by others and experiencing social and natural connectedness can support people with mental health problems to integrate socially and improve their communication skills through shared reflection [19,23,50,96]. Therefore, some of the benefits of nature-based therapy might arise from it increasing social cohesion. Additionally, this might be a transfer effect, since the nature connection improved and through this, the general ability to connect with other people as well. Through the different nature-based therapy sessions, it was tremendous to see the patients grow as well as the plants. Most of them were quiet in the beginning, withdrawn like a seed in the ground. However, with more time in the rehabilitation process, they started being more outgoing, and they started to sprout and grow. They asked for activities to start, chose themselves what they wanted to do in the garden, and found the individual projects they were happy to work on—started blooming. They found motivation in doing gardening work, even if they did not like it from the beginning. One client said, “I would never have thought so, but garden activities are really great, and I like doing them.” These observations from the researcher are congruent with the findings from Pálsdóttir et al. [21]. The researcher found that initially the clients are more withdrawn, and later on show active participation. This effect might be explained by a quote from Berger: “We never look at just one thing; we are always looking at the relation between things and ourselves” [98]. He describes that nature might have served as a mirror for patients during nature-based therapy. One patient said that the cycles from nature reminded them that they also need time for themselves to retreat. These results are also found in other studies, where events in nature are interpreted symbolically by participants in nature-based therapy and lead to reflections on their own choices in life [20,21]. Moreover, results from the participants’ self-assessment questionnaire on the effectiveness of nature-based therapy showed that they considered the therapy to be effective in learning how to cope with problems and illness (see Figure 5). A main theme in the qualitative interviews was also that the participants experienced a higher self-efficacy and self-acceptance through nature-based intervention. The nature-based therapist was highly praised for being empathetic, and reciting interesting poems, and can be seen to have been an eye opener for many. Patients experienced this as valuable since they discovered new interests, experienced self-efficiency, and became inspired to change. One patient said, “I realized that I can be a creator. Through sowing seeds, I realized that I can create and nurture new things with my hands. Influence other lives”. Thus, self-efficiency and self-acceptance might lead to a positive mindset, which is associated with health and well-being (lower levels of inflammation) [99]. This is consistent with the findings from Pálsdóttir, Grahn, and Persson [88]. In their research, nature-based rehabilitation led to a positive change in their perceived values of everyday occupation, inspiration for transformation, and a more sustainable lifestyle. Additionally, being outside and being exposed to more light/brightness might lead to increased self-awareness [32]. Through psychoeducation (part of the integrative therapy program), the clients learn mindfulness practice, which might be an influencing factor in the nature-based therapy as well. The participants learned to be present in the here and now, and thus being able to experience the qualities of nature. This has been shown to be effective in other therapy settings [59,64,65,100]. It is interesting to look at the finding that everyone felt an improvement on the soul level through nature-based therapy (see Figure 5). To our knowledge, no studies have investigated an improvement on people’s soul level in nature-based therapy yet. However, attention has been paid to the connection between people’s soul, their health, and being in nature and gardens [101]. It is unclear what each patient understood as ‘soul improvement.’ This study makes a connection between spirituality and soul, and we refer to the definition by Brown, Carney, Parrish, and Klem as “[…] a sense of connectedness to a higher power and openness to the infinite beyond human existence and experience” [102] (pp. 110–111). Research investigating spirituality in mental health found a reduction in mental and emotional illness in individuals with high spirituality [102]. The World Health Organization has discussed the existential dimension of health, which they define to be based on eight components: spiritual connection, meaning and purpose in life, the experience of awe and wonder, wholeness and integration, spiritual strength, inner peace, hope and optimism, and faith [103]. Melder [103] found that people’s contact with nature was one of four factors most often mentioned as meaning-making by the subjects, and this can be linked to a stronger existential dimension in life [103]. An explanation might be the biophilia hypothesis, which states that human beings come from nature, have an urge to be in nature, and to (re-) connect to nature [97]. Furthermore, this might be explained by the findings of Sahlin et al. [104], namely that activities in the garden allowed patients to practice doing one thing at a time, not rushing through things, and allowing oneself to take breaks [104]. Moreover, belonging to a social context, since the patients were with other patients with similar diagnoses in the same therapy program and sessions, might have been experienced as supportive, too. Additionally, the transdisciplinary therapist team with the holistic approach might have been vital for developing strategies and tools to better face and manage everyday demands. Altogether, this might lead to an improvement on the soul level—a holistic approach and understanding oneself through connection with nature and others and having a place in the world.

Something we need to highlight is the importance of the therapist. Sometimes a skilled therapist is needed to help in the rehabilitation process, to be a catalyst, and open one’s eyes, as the participants sometimes put it. Sahlin et al. [104,105] mention that the botanist in the rehabilitation garden of Gröna Rehab in Gothenburg allows the participants to discover nature during walks in the nearby nature reserve. Each participant has their own magnifying glass to help them. The botanist invites participants to use it and discover through the magnifying glass the nuances and riches hidden in flowers, mosses, butterflies, etc. They experience a grandeur, awe, and wonder in nature that touches them deeply and involves their own situation on an existential level. It includes a forgiving attitude toward both the environment and themselves, gives them hope, and starts new coping strategies [104,105]. In the same way, the therapist in this study helps the participants to look at nature in a new way, such as by interpreting annual rings symbolically. Experiencing the greatness and beauty, being deeply touched, and feeling trust in nature, an awe, a connection, and perhaps love, can, according to some controlled studies, give the participants strength and courage, as well as give new perspectives on themselves and existence [106,107]. Oxytocin is sometimes called the hormone of love, and the Calm and Connection Theory suggests that the oxytocinergic system is activated in a very powerful way when a person feels security, attraction to, and connectedness to natural environments, which affects the individual on a deeper level [50]. Accordingly, this awe, attraction to, and perhaps the love of the natural environment can, mediated by the oxytocinergic system, lead the persons to transcend their way of being and thinking, about both their selves and existence. It can lead to a change in their internal working model (that is, their routine ways of feeling, thinking, and acting), making them understand how to deal with life. This, in turn, leads to improvements in one’s physical and mental health [91,108,109].

With our third hypothesis, we intended to investigate whether there is a difference in the improvement of psychological well-being in patients with low or high connectedness to nature after nature-based therapy. Does connectedness to nature improve mental well-being? The question as to whether a nature lover (means higher connectedness to nature at the beginning of the therapy) recovers better from nature-based therapy remains unanswered. No statistically significant differences were found in the comparison. Connectedness to nature had no influence in predicting WOHL in the linear mixed effect analysis. Thus, there was no evidence for the third hypothesis, namely that there would be a difference in the improvement between individuals scoring low/high on connectedness to nature through nature-based therapy. These results might be due to a very small sample size, resulting in low statistical power. Nevertheless, a trend is visible when comparing WOHL in the baseline and post-testing. This could be investigated through further research. It is possible that individuals scoring low on connectedness to nature have a bigger chance of improvement than the ones already scoring high on connectedness to nature.

More females (13–16, ≙ 81–84%) than males (3 ≙ 15–18%) participated in the two measurements (baseline and post-testing) (see Table 1). This is consistent with the disease contribution among gender. More females than males have a diagnosis of depression [7].

The frequency of participation in nature-based therapy might have influenced the outcome (dose-respond relationship). Since individual psychotherapy sessions took place during nature-based therapy, sometimes patients could not attend nature-based therapy. Furthermore, the length of stay varied among the patients (some stayed four, some up to six weeks). Nevertheless, the frequency of attendance was recorded per patient. This was evaluated, revealing that there were no big differences in the participation of nature-based therapy from patients during their stay. They attended between 9–12 sessions during their stay.

It is essential to focus on young individuals’ health and well-being as the older they get, the more established (unhealthy) their routines become, resulting in a lifestyle that may be unhealthy [10]. Additionally, the costs of healthcare increase if those with long-term problems are not treated directly [9]. Moreover, in young adults, severe depression can lead to suicide, the fourth leading cause of death in 15–29year-olds [5]. Helping individuals at an early stage of depression might lead to a more fulfilled life for them and may lead to an uplifting snowball effect for their (social) surrounding. Furthermore, healthcare costs might stay low when the root cause of a problem is treated directly [9]. Consequently, less time might be needed to treat the other side effects that may be leading to unhealthy habits.

### Limitations

This study gives the first insights into nature-based therapy within the integrative therapy program at the GKH for psychosomatic patients, but many questions remain unanswered.

This study is a pilot study with a pre/post-testing design without a control group. This study design was deliberately chosen for a pilot study because it is not possible and reasonable to separate nature-based therapy as a single element at this stage. Due to the limited design of the study, it is not possible to highlight the nature-based therapy and its independent effect without the other therapy elements of the integrative therapy program. Furthermore, many other confounders can explain the effects found. In this case, to investigate an apparent effect of the nature-based therapy, the control group would receive all parts except the nature-based therapy. Andrews [110] maintained that the randomized controlled trial design is demanding to use in rehabilitation research and that other strategies can provide equally safe or safer results. In addition, Graham et al. [111] suggest smaller projects with a clear focus on individual participants as a good research strategy in rehabilitation research. Because they have many benefits, it has been recommended to use quasi-natural experiments more often in rehabilitation research. Among other things, ethical problems are reduced, and the result gives a fairer picture of how real rehabilitation works [112,113]. It would, therefore, be useful to conduct further studies with a more sophisticated study design including a control group, preferably as natural experiments as a form of randomization. The weather influenced the nature-based intervention. During the time of data collection, through mixed methods from May until August, the temperature was warm, with mainly sunny days during the therapy. It remains unclear how the weather influenced the outcome of this research. Further investigations in that direction are needed in future research.

Every person’s perspective with regard to their individual, cultural and geographical backgrounds, including mental state of mind, physical ability, emotional awareness and regulation ability, gender, family background, socio-economic factors, character (e.g., extrovert vs. introvert), as well as possible stigma for the disorders and individual’s spirituality influences the nature-based therapy. Additionally, past experiences with gardens, forests, and nature, and time spent in nature (before and after the therapy) have to be considered. The influencing factor of each person being an individual and therefore the group climate must be considered. The group climate questionnaire [114] could be used in further research to reveal more insights into the group processes and dynamics in group therapy at work. The time an individual spent at a place and the place attachment that occurs should be considered to influence the outcome, too. Moreover, investigating how the participants use the garden (also on their own, when there is no therapy) would give more insights into the processes (place attachment) and understanding who experienced what as a supportive environment.

The influence of the therapist, the offered activities (the therapy program), as well as the design and organization of the garden, the season, and the time during the day when the nature-based therapy takes place, have to be considered. These circumstances make it particularly difficult to compare the different nature-based therapy sessions and programs.

It is suggested that further studies on nature-based therapy have a longer duration [114] and investigate the underlying influencing factors from connectedness to nature (the feeling of connectedness within the group, spirituality, and meaningfulness of the activity in the garden).

Further research could also investigate the different qualities of the two places (the forest and the garden) of the nature-based therapy taking place at the GKH. Research suggests that both places have different qualities, and the combination is suggested to be promising for therapy outcomes [89,115,116]. Nevertheless, these are expectations that might apply to the places at GKH as well. The perception of these places needs further evaluation in further research with, for example, the PSD [21], the evaluation tool suggested by Bengtsson and Grahn [25] or Tudaor [117], since currently, there is no evaluation from the clients’ perspective at the GKH.

Nature-based therapy might have fostered the transfer effect from the clinic to everyday life. The qualitative results revealed that some patients took a trophy home, a plant, a tree pit, and a poem from the therapist. This could lead to a longitudinal positive effect on mental health in patients and should be investigated in further studies via a follow-up study design. Additionally, this could provide further explanation on how sustainable/stable these effects of mental health and connectedness to nature are. Did the transfer of the integrative therapy program into everyday life happen? A longitudinal multiple assessment design with follow-up assessments is suggested (e.g., one baseline before attending the intervention, a follow-up at the GKH, and one assessment after the last intervention at the GKH, and follow-up assessments at home after one month, six and 12 months after the intervention—to also be able to investigate the long-term effects of the therapy approach). A control group without nature-based therapy but including all other therapy elements is also suggested.

No gender-specific results could be revealed since mainly women participated in this research, and one trans person (not visible in the gender questionnaire).

## 5. Conclusions

This research gives insight into the influences of the integrative therapy program with a focus on nature-based therapy on the health and well-being of psychosomatic patients. The aim was to better understand how connectedness to nature influences mental well-being and to develop nature-based therapy programs. Another aim was to promote the health effects that these little garden spaces have on patients. Indeed, the results of this study indicate that nature-based therapy as part of the integrative treatment can be an effective treatment for psychosomatic disorders. Furthermore, the results show that connectedness to nature and mental well-being are connected in psychosomatic patients, and both significantly improve over time. Effectiveness, satisfaction, and improvements on the mood and soul level, well-being, contact with other people, and coping with problems and ill health from the patients’ perspective are reported.

To conclude, this data contributes to the positive effect of integrative therapy concepts and that it is possibly influenced by nature-based therapy. Moreover, it highlights the need for such therapy to be a publicly funded service, accessible and beneficial for all. To our current knowledge, no studies were found that investigated connectedness to nature in the German population, nor in the context of nature-based intervention in psychosomatic patients. To promote the conclusions from this study, further research, for example, with a mixed methodology (interviews as a qualitative approach), a bigger sample size, a control group (RCT), and a longer period of time (ideally, several years), with focus on a cumulative, specific investigation on all therapy elements from the integrative therapy, is needed. Especially, it is important to find out who benefits the most from nature-based therapy (patients with initially low/high connectedness to nature). Moreover, the qualities of the natural environment where the nature-based therapy takes place have to be taken into account from the patients’ perspective as well.

## Figures and Tables

**Figure 1 ijerph-20-02167-f001:**
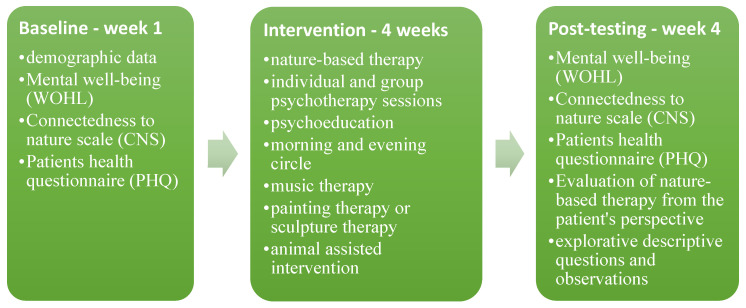
Visualization of the study design about how research was being conducted (including measurement tools, integrative therapy program, and time of the research).

**Figure 2 ijerph-20-02167-f002:**
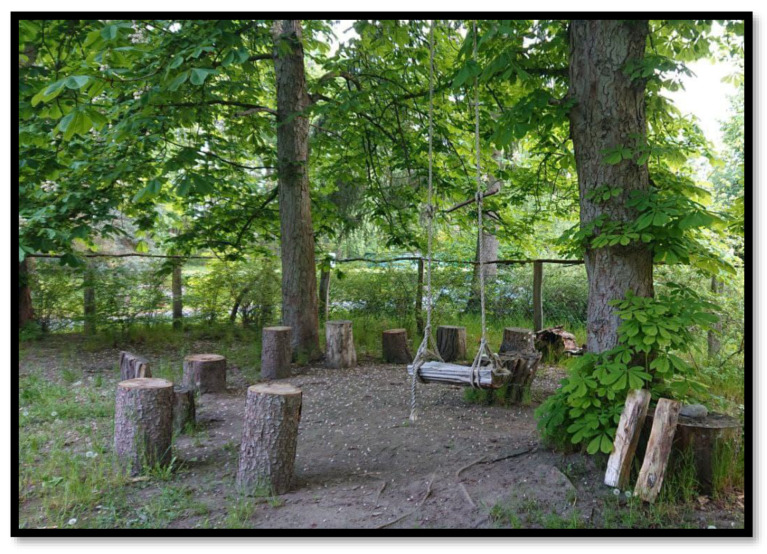
The circle of logs where each nature-based therapy session starts is in the back of the garden. Image by: Lilly Joschko ©.

**Figure 3 ijerph-20-02167-f003:**
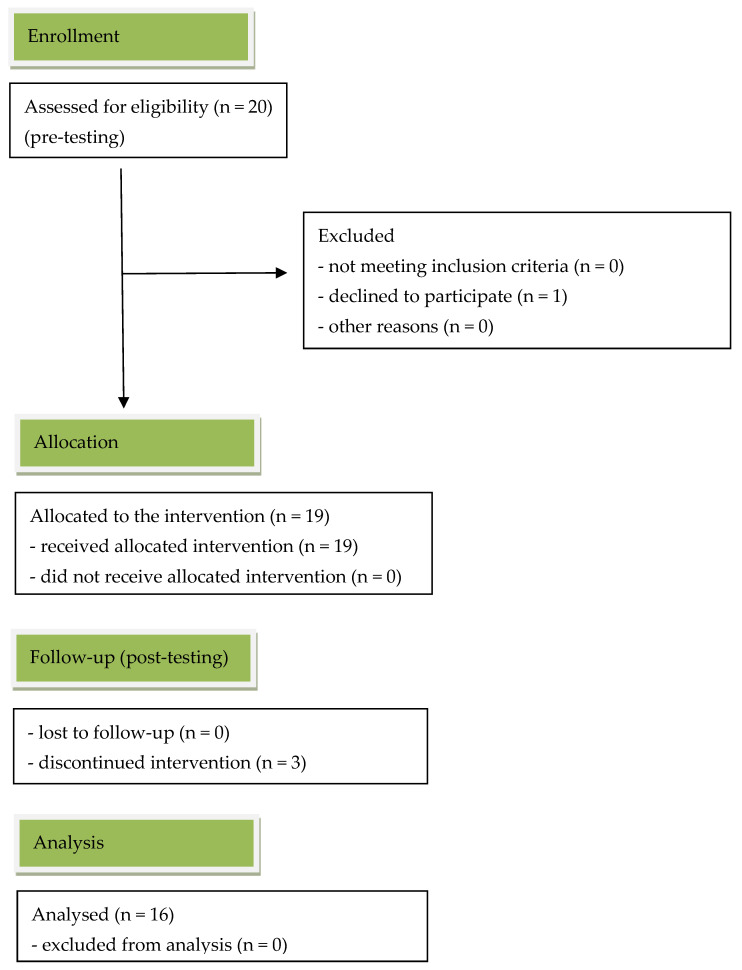
CONSORT flow-diagram: transparent records of participants for the study.

**Figure 4 ijerph-20-02167-f004:**
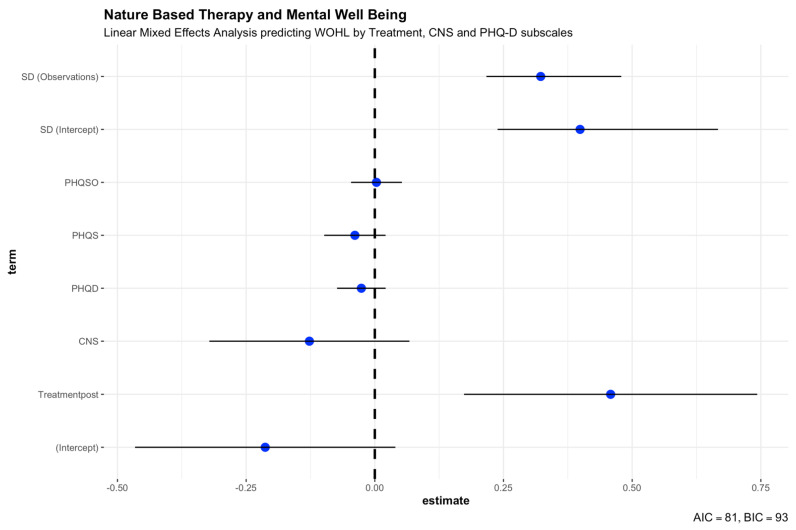
Linear mixed effects analysis. Model 2 predicting mental well-being (WOHL) by Treatment, Connectedness to Nature (CNS), and PHQ-D subscales for depression, somatoform disorders, and stress.

**Figure 5 ijerph-20-02167-f005:**
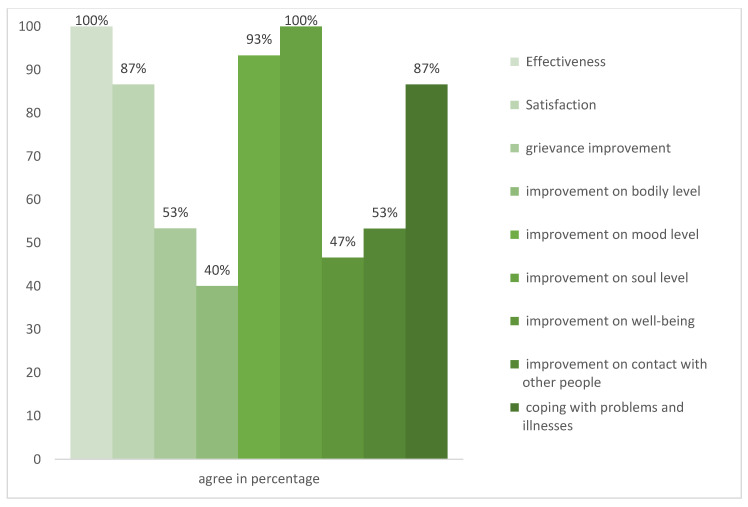
Experienced effectiveness, satisfaction, and improvements from nature-based therapy based on the self-rated post-testing questionnaire.

**Table 1 ijerph-20-02167-t001:** Demographic data (age and gender) of the patients divided into the baseline and post-testing conditions.

Condition	n	Age Range	Mean	Gender	
				Male	Female
Baseline	19	18–27	21.32	3 (15.3%)	16 (84.2%)
Post-testing	16	18–27	21.5	3 (18.75%)	13 (81.25%)

**Table 2 ijerph-20-02167-t002:** Paired samples *t*-test of the baseline and post-testing comparison.

Outcome	Baseline Mean (SD)	Post-Testing Mean (SD)	Diff.	95% Confidence Interval of the Difference		*t*	df	Sig. (2-Tailed)
				Lower	Upper			
WOHL	1.05 (0.44)	1.60 (0.57)	0.55	0.83	0.27	−4.16	15	0.001
CNS	3.91 (0.76)	4.37 (1.01)	0.45	0.77	0.12	−2.94	15	0.010
PHQ-9	16.94 (4.96)	12.94 (4.34)	3.31	0.10	6.52	2.20	15	0.044
PHQ stress	8.94 (3.79)	7.12 (3.72)	1.81	−0.81	4.44	1.47	15	0.162
PHQ somatoform	11.81 (5.79)	11.00 (4.82)	0.81	−2.56	4.18	0.51	15	0.615

**Table 3 ijerph-20-02167-t003:** Linear Mixed Effects Analysis. Model 2 predicting mental well-being (WOHL) by Treatment, Connectedness to Nature Scale (CNS), and PHQ-D subscales (PHQ-9, PHQ-15, and stress).

Model 2 Summary							
Parameter	Co-Efficient	95% CI	t (27)	*p*	Std. Co-ef.	Std. Co-ef. 95% CI	Fit
(Intercept)	−0.21	(−0.47, 0.04)	−1.73	0.095	−0.36	(−0.78, 0.07)	
Treatment (post)	0.46	(0.17, 0.74)	3.30	0.003	0.77	(0.29, 1.24)	
CNS	−0.13	(−0.32, 0.07)	−1.34	0.191	−0.22	(−0.56, 0.12)	
PHQ-9	−0.03	(−0.07, 0.02)	−1.14	0.265	−0.22	(−0.61, 0.17)	
PHQ stress	−0.04	(−0.10, 0.02)	−1.33	0.194	−0.25	(−0.64, 0.14)	
PHQ-15	0.00	(−0.05, 0.05)	0.13	0.895	0.03	(−0.42, 0.48)	
Patients‘ ID—random	0.40	(0.24, 0.67)					
Residual—random	0.32	(0.22, 0.48)					
AIC							80.95
BIC							93.39
R^2^ (conditional)							0.74
R^2^ (marginal)							0.35
Sigma							0.32

Random effect: patients’ ID. Fixed effects: Treatment, Connectedness to Nature Scale (CNS), Depression (PHQ-9), Stress (PHQ-D), and Somatoform Disorders (PHQ-15). Standardized parameters were obtained by fitting the model on a standardized version of the dataset. 95% Confidence Intervals (CIs) and *p*-values were computed using a Wald t-distribution approximation.

## Data Availability

Quantitative data are available by contacting the corresponding author.

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
