# Peer review of "Nature-Based Therapy in Individuals with Mental Health Disorders, with a Focus on Mental Well-Being and Connectedness to Nature—A Pilot Study"

_ijerph, 2023, doi:10.3390/ijerph20032167_

Round 1
Reviewer 1 Report
Overall very interesting paper and I value the insight regarding a young population. I can imagine drawing on this paper for my own scholarship. With this said, I will suggest some edits. The abstract could have clearer reference to the links to the pandemic and lockdowns. Arguments for nature based therapy as linked to biopsychosocial models is solid. "Soul Level" is used in the abstract (line 20). IF this term is used it must me clearly defined in the paper. Line 238 "wisdom" of the therapist should be explained. The qualitative Data analysis process should be clarified. Was it a thematic analysis or was another method used. How did the authors arrive at the theme. Was there a coding process that help identify the themes? I would also suggest a clearer methodology beyond qualitative. Perhaps, descriptive exploratory methodology which is a broad qualitative methodology. The quantitative methodology and methods are clearly articulated.
Author Response
Reviewer 1
Comments and Suggestions for Authors:
Overall very interesting paper and I value the insight regarding a young population. I can imagine drawing on this paper for my own scholarship. With this said, I will suggest some edits.
- The abstract could have clearer reference to the links to the pandemic and lockdowns. Arguments for nature based therapy as linked to biopsychosocial models is solid.
ANSWER: We added a subsentence to the abstract to further clarify the connection to the pandemic.
- "Soul Level" is used in the abstract (line 20). IF this term is used it must me clearly defined in the paper.
ANSWER: Thank you for this note. “Soul level” can be defined as a function of a person's being, presence or consciousness. We have added that definition at page 8, Line 397, together with references. Also, we discuss soul level in Discussion, page 18. Therefore, we should mention improvement in soul level in the Abstract.
- Line 238 "wisdom" of the therapist should be explained.
ANSWER: Good point. We added the subsentence of his own coping strategies to make sure what is meant by wisdom. (p.5, line 271)
- The qualitative Data analysis process should be clarified. Was it a thematic analysis or was another method used. How did the authors arrive at the theme. Was there a coding process that help identify the themes?
ANSWER: Thanks for asking for the themes. We have added how we came to the themes and the literature for that (p.9, line 455f).
- I would also suggest a clearer methodology beyond qualitative. Perhaps, descriptive exploratory methodology which is a broad qualitative methodology.
ANSWER: In Figure 1 post-testing we name it “explorative descriptive questions and observations”. True that we don’t have it in the text yet. We added it at line 211, p.4.
- The quantitative methodology and methods are clearly articulated.
ANSWER: Thank you very much for your helpful comments and valuation of the research.
Reviewer 2 Report
I don't remember when I had previously read such an inspired exposition of the romance of psychotherapy. And for this I express my great gratitude to the authors. I will also note the competent application of methods of applied statistics, including such a modern and complex method as "linear mixed models".
However there is one weighty objection. The study is not experimental – a pre–experimental design is used (preliminary testing – impact - final testing) with one group, without any control group. Thus, it is impossible to separate the effect of therapy from many other competitive, extraneous factors, such as background influence, natural development, the effect of preliminary testing, etc. (Campbell D.T., Stanley J.C., 1976). At this stage, this objection is unavoidable.
Author Response
Reviewer 2
Comments and Suggestions for Authors
I don't remember when I had previously read such an inspired exposition of the romance of psychotherapy. And for this I express my great gratitude to the authors. I will also note the competent application of methods of applied statistics, including such a modern and complex method as "linear mixed models".
- However there is one weighty objection. The study is not experimental – a pre–experimental design is used (preliminary testing – impact - final testing) with one group, without any control group. Thus, it is impossible to separate the effect of therapy from many other competitive, extraneous factors, such as background influence, natural development, the effect of preliminary testing, etc. (Campbell D.T., Stanley J.C., 1976). At this stage, this objection is unavoidable.
ANSWER: This point is correct. It would have been nice if it had been possible to conduct a form of RCT study, but we had neither the financial nor the structural possibilities to do so in the current study. For a small pilot study, we found it appropriate to conduct a small exploratory study without a control group under the given structural conditions in the ongoing regular care operation. To address this objection, we added a whole paragraph to highlight this limitation in the Limitations section on page 19 from line 948ff.
Reviewer 3 Report
- I congratulate the authors of this study. It is very interesting, novel and without a doubt, necessary.
- Among the strong points, I want to highlight that a new and holistic type of therapy has been addressed.
- It is also a strong point that two types of analysis have been carried out, one quantitative and the other qualitative.
SUGGESTIONS FOR IMPROVEMENT / DOUBTS:
- The article is too long. I think the extension can be reduced.
Example/suggestion. Perhaps the paragraph describing the characteristics of depression according to the WHO (lines 40-44) can be deleted. I do not think that it's necessary.
And given that all the data appear in the tables, perhaps in the Results section it is not necessary to include all the data, percentages, all the statistics, but rather reference can be made to the results that come out significant and to those that do not come out, and to see the specific data, refer to the tables, which are clearly illustrative.
- Excessive bibliography. 110 Bibliographic references are excessive. Maybe it can be cut in half.
- I do not understand what the phrase that appears in lines 399-400 means (The participants can either be referred by their medical professional or can apply themselves to stay at the GKH hospital). Are the patients participating in the study all admitted to the hospital? Or, some are referred by their medical professional and others were admitted to the hospital?
- Is the garden where the contact therapy with nature takes place, is it the hospital garden? It is not clearly specified.
- In relation to the sessions in the forest, is the forest close? do you have to travel by car? What kind of activity is done in the forest? walk/walk, observe vegetation/animals?
- The improvement on soul level, is a somewhat subjective term and, as the authors indicate, we cannot be sure of what each user could understand. I find it a bit shocking. The term spirituality is more widespread and there are numerous studies that address it. The term soul level is a bit problematic for me.
- An element that confuses me a bit is about the type of patients/users. On the one hand, it indicates that they are patients with psychosomatic problems (by which I understand that they have a physical illness, even if it is affected by psychological circumstances/variables). And on the other hand, especially in the Discussion section, there has been talk of depression, that therapy has positively improved depression. This is a bit confusing. Did all patients have depression? Did the therapy only improve the psychological aspects –depression- or also the more physical part –psychosomatic illness-?
- Regarding the limitations of the study, there is nothing to object to since, in the Limitations section, the authors have been honest when collecting numerous variables that should be taken into account and that may affect the results. On the other hand, I understand that some of the questions that have not been answered are future lines of investigation to advance in a deeper knowledge.
Author Response
Reviewer 3
Comments and Suggestions for Authors
- I congratulate the authors of this study. It is very interesting, novel and without a doubt, necessary.
- Among the strong points, I want to highlight that a new and holistic type of therapy has been addressed.
- It is also a strong point that two types of analysis have been carried out, one quantitative and the other qualitative.
SUGGESTIONS FOR IMPROVEMENT / DOUBTS:
- - The article is too long. I think the extension can be reduced.
Example/suggestion. Perhaps the paragraph describing the characteristics of depression according to the WHO (lines 40-44) can be deleted. I do not think that it's necessary.
ANSWER: We understand your point of shortening the article. But since it’s a transdiciplinary paper and should be understand by everyone also outside of the field we would like to keep the WHO depression classification/definition. I would not assume everyone who read the article to know the definition of depression. We check if we find other paragraphs/sentences that we can shorten. We attempted to shorten additional paragraphs in several other places in the manuscript (see manuscript in revision mode).
- And given that all the data appear in the tables, perhaps in the Results section it is not necessary to include all the data, percentages, all the statistics, but rather reference can be made to the results that come out significant and to those that do not come out, and to see the specific data, refer to the tables, which are clearly illustrative.
ANSWER: Many thanks for the suggestion. We have revised the results throughout and shortened the data as far as possible where they are available in tables (see Results section).
- - Excessive bibliography. 110 Bibliographic references are excessive. Maybe it can be cut in half.
ANSWER: We tried to delete some references throughout the manuscript, but we were not able to cut it by half. We have three other reviewers who express great appreciation for the article. Therefore, we must be very careful when cutting the article and references. We revised certain things, such as long descriptions of the characteristics of depression and repeating all the data, percentages and statistics that can be read from tables.
- - I do not understand what the phrase that appears in lines 399-400 means (The participants can either be referred by their medical professional or can apply themselves to stay at the GKH hospital). Are the patients participating in the study all admitted to the hospital? Or, some are referred by their medical professional and others were admitted to the hospital?
ANSWER: Patients may be admitted by their general practitioner or may self-refer to the hospital for psychotherapy treatment. We have revised the sentence in the manuscript to hopefully make it more understandable (see p.9, line 486).
- Is the garden where the contact therapy with nature takes place, is it the hospital garden? It is not clearly specified.
ANSWER: The therapy garden of the GKH is located on the hospital grounds at a distance of about 150 meters from the patients' house. We added this sentence to explain the location of the garden in the manuscript under “procedure”page 5 line 258ff.
- In relation to the sessions in the forest, is the forest close? do you have to travel by car? What kind of activity is done in the forest? walk/walk, observe vegetation/animals?
ANSWER: The hospital is partly surrounded by forest in which the forest therapy units take place. We added a small paragraph about what is done during nbt in the forest (p.5, line 277ff).
- The improvement on soul level, is a somewhat subjective term and, as the authors indicate, we cannot be sure of what each user could understand. I find it a bit shocking. The term spirituality is more widespread and there are numerous studies that address it. The term soul level is a bit problematic for me.
ANSWER: The term soul level is used by GKH for all interventions. “Soul level” can be defined as a function of a person's being, presence or consciousness. We have added that definition at page 8, line 398, together with some references. Also, we discuss soul level in Discussion, page 18.
- An element that confuses me a bit is about the type of patients/users. On the one hand, it indicates that they are patients with psychosomatic problems (by which I understand that they have a physical illness, even if it is affected by psychological circumstances/variables). And on the other hand, especially in the Discussion section, there has been talk of depression, that therapy has positively improved depression. This is a bit confusing. Did all patients have depression? Did the therapy only improve the psychological aspects –depression- or also the more physical part –psychosomatic illness?
ANSWER: All patients participating in this study received a diagnosis of depression (mild to severe). Many patients had additional diagnosis. For example: somatoform disorders, insomnia, acrophobia, anxiety disorders, social phobia, post-traumatic stress disorder (PTSD), anorexia, obsessive-compulsive disorders, panic disorder, and bulimia (see Results – sample description on page 9). Due to the variety of different diagnoses, it is not possible to examine the improvement through therapy for each diagnosis. Therefore, we focused on selected outcomes like mental wellbeing (WOHL from the Health-49) and on the other outcomes like depression and stress from PHQ.
- Regarding the limitations of the study, there is nothing to object to since, in the Limitations section, the authors have been honest when collecting numerous variables that should be taken into account and that may affect the results. On the other hand, I understand that some of the questions that have not been answered are future lines of investigation to advance in a deeper knowledge.
ANSWER: To report it more clearly, we added an additional paragraph on the limitation of a missing control group and randomization, prompted by a comment from Reviewer 2 (see Limitations section, p19).
Reviewer 4 Report
Dear authors
I thoroughly enjoyed reading your manuscript. I believe it reads very well, is highly policy relevant, important to this area of research, and it is presented in a very well-structured manner.
You chose an interesting and well-thought-out research strategy, and used mixed methods, to investigate the relationship between nature-based therapy, mental health, and individuals’ connectedness to nature. You conclude that your study gives a first insight into the processes of nature-based therapy and its effectiveness. You recommend future research that would build on and complement your findings from this pilot study.
Basic reporting (I have entered specific comments in the attached file)
· The language used throughout is clear, concise, and easy to understand.
· The introduction is clear and well referenced. It explains well the need for this research and the strengths of the chosen research strategy and mixed methods.
· The figures used are generally relevant and clear and support the text well. I would have liked more photos of the garden/therapy area, to get a more clear and complete understanding of the environment and atmosphere.
· The applied methods are described very well, and so it should be possible to follow and repeat the procedure for future research. Again, photos of the outdoor areas would be helpful.
· The data is interpreted appropriately and consistently and the process is explained well.
Suggestions for consideration
Introduction: At the end of the introduction, you state your objectives and hypotheses. It reads as if you objectives are the same as you hypotheses. I would suggest writing these clearly and separately, in list form, to make it easier to read.
Hypothesis 1......
Hypothesis 2......
Hypothesis 3......
Also, I would recommend writing it more clearly what your primary research question is.
Discussion - Line 577. You write:……. These changes might be due to the treatment the patients received in the nature, but it might also be a side effect from the other therapy sessions and activities that clients received.
- This is an important point and a limitation. Were the participants questioned about all the other elements of the very complex treatment? Did they find those elements effective also? How would you know if the nature-based therapy would have worked in isolation, without all the other elements? I understand the aim of your study was not to evaluate the whole treatment and its complexity, but further justification/discussion relating to these issues would be welcomed.
- I would suggest referring back to hypothesis 1, 2 and 3 one at the time, so there is a direct link back to the introduction and it is clear you have addressed everything you set out to investigate
Conclusion:
- I would suggest the importance of looking at nature-based therapy in isolation, but also specifically investigate the synergistic/cumulative effect between the different treatment elements as presented in your study.
- You didn't investigate any of the other therapies included in the treatment. Therefore, there is a possibility that the nature-based therapy would have been successful without any of the other treatments.
In your discussion and conclusion, you have to be a bit careful with the wording; in places it reads as if you are concluding on things you haven’t actually investigated.
I wish you good luck with publication of a great manuscript.

Author Response
Reviewer 4
Comments and Suggestions for Authors
Dear authors
I thoroughly enjoyed reading your manuscript. I believe it reads very well, is highly policy relevant, important to this area of research, and it is presented in a very well-structured manner.
You chose an interesting and well-thought-out research strategy, and used mixed methods, to investigate the relationship between nature-based therapy, mental health, and individuals’ connectedness to nature. You conclude that your study gives a first insight into the processes of nature-based therapy and its effectiveness. You recommend future research that would build on and complement your findings from this pilot study.
Basic reporting (I have entered specific comments in the attached file)
ANSWER: We looked at the attached file. We addressed these and tried to integrate these comments with your suggestions below.
All changes are made in the manuscript with track changes. We hope that these fulfill your suggestions.
- The language used throughout is clear, concise, and easy to understand.
- The introduction is clear and well referenced. It explains well the need for this research and the strengths of the chosen research strategy and mixed methods.
- The figures used are generally relevant and clear and support the text well. I would have liked more photos of the garden/therapy area, to get a more clear and complete understanding of the environment and atmosphere.
- The applied methods are described very well, and so it should be possible to follow and repeat the procedure for future research. Again, photos of the outdoor areas would be helpful.
- The data is interpreted appropriately and consistently and the process is explained well.
ANSWER: Thank you very much for your feedback.
Suggestions for consideration
- Introduction: At the end of the introduction, you state your objectives and hypotheses. It reads as if you objectives are the same as you hypotheses. I would suggest writing these clearly and separately, in list form, to make it easier to read.
Hypothesis 1......
Hypothesis 2......
Hypothesis 3......
Also, I would recommend writing it more clearly what your primary research question is.
ANSWER: We revised the paragraph with the three hypotheses and rewrote the research question hopefully more clearly on page 3 in the last paragraph of the introduction.
- Discussion - Line 577. You write:……. These changes might be due to the treatment the patients received in the nature, but it might also be a side effect from the other therapy sessions and activities that clients received.
- This is an important point and a limitation. Were the participants questioned about all the other elements of the very complex treatment? Did they find those elements effective also? How would you know if the nature-based therapy would have worked in isolation, without all the other elements? I understand the aim of your study was not to evaluate the whole treatment and its complexity, but further justification/discussion relating to these issues would be welcomed.
ANSWER: Thank you for your valuable comments. We revised the part on p16, line 730 and we added a whole paragraph on page 19 at the beginning of the limitations section to further address the issue that there is no direct, isolated result for the effect of nature-based therapy in our study due to the absence of a control group and missing randomization. In this case, to investigate a clear effect of the nature-based therapy, the control group would receive all elements except the nature-based therapy. However, this study was part of a broader study evaluating the whole therapy program from patients’ perspective. Thus, there is also data about patients’ preference of each therapy element. Unfortunately, this data would have been too much for this paper and is also in an ongoing investigation.
- I would suggest referring back to hypothesis 1, 2 and 3 one at the time, so there is a direct link back to the introduction and it is clear you have addressed everything you set out to investigate
ANSWER: We tried to write more clearly when a paragraph is beginning to address a hypothesis and which hypothesis (see Discussion section).
- Conclusion:
- I would suggest the importance of looking at nature-based therapy in isolation, but also specifically investigate the synergistic/cumulative effect between the different treatment elements as presented in your study.
- You didn't investigate any of the other therapies included in the treatment. Therefore, there is a possibility that the nature-based therapy would have been successful without any of the other treatments.
ANSWER You are right. As stated above under point 2, we added another paragraph addressing these issues. We hope that this makes it more precise.
- In your discussion and conclusion, you have to be a bit careful with the wording; in places it reads as if you are concluding on things you haven’t actually investigated.
ANSWER: Good point. We read the discussion and conclusion again and revise it.
- I wish you good luck with publication of a great manuscript.
ANSWER: Thank you very much
Round 2
Reviewer 2 Report
The alterations made to the manuscript (especially in the Limitations section) explain the shortcomings of the study design. But they do not eliminate them.